# A Comparative Study of the Anatomy of Leaf Domatia in *Gardenia thunbergia* Thunb., *Rothmannia capensis* Thunb., and *Rothmannia globosa* (Hochst.) Keay (Rubiaceae)

**DOI:** 10.3390/plants11223126

**Published:** 2022-11-16

**Authors:** Sivuyisiwe Situngu, Nigel P. Barker

**Affiliations:** 1School of Animal, Plant and Environmental Sciences, University of the Witwatersrand, Johannesburg 2001, South Africa; 2Department of Plant and Soil Sciences, University of Pretoria, Private Bag X20, Hatfield, Pretoria 0028, South Africa

**Keywords:** domatial anatomy, cuticular folds, plant–mite mutualism, transmission electron microscopy, light microscopy

## Abstract

Many dicotyledonous plants produce structures called leaf domatia. Approximately 28% of 290 families have species with leaf domatia. These structures are abundant within the Rubiaceae and Vitaceae. 26% and 16% out of 206 representative species cited in literature from 48 plant families belong to the Rubiaceae and Vitaceae respectively. Leaf domatia are usually associated with mites and often mediate mutualistic relationships with predacious mites. These structures are pockets found in the underside of the leaf, where the secondary vein axils meet the major vein. In the present study, we examine the anatomical structures of leaf domatia from three plant species (*Gardenia thunbergia* Thunb., *Rothmannia capensis* Thunb., *Rothmannia globosa* (Hochst.) Keay) from the Rubiaceae family in order to find out if their internal tissues differ. These plants were sectioned and viewed under a Light Microscope in order to document their internal anatomy. A Transmission Electron Microscope was used to search for the presence of cuticular folds in their epidermis, which are thought to assist plant to communicate with mites. Results from this study suggested that the main features of domatial anatomy are the presence of an extra layer of tissue in the lower epidermis, a cuticle, cuticular folds, trichomes and the presence of an invagination. Cuticular folds were present inside the domatia but were not restricted to the domatial lamina. Thus, we conclude that these structures do not assist plant in plant-mite communication. This study provides a better understating of the anatomy of leaf domatia of the Rubiaceae.

## 1. Introduction

Leaf domatia are structures found in the axils of major veins on the abaxial side of leaves, and sometimes on the bases of leaflets in compound leaves (Figure 1). These are common features within dicotyledonous families such as the Rubiaceae, Vitaceae, Annonacea, and Bignoniaceae, to name but a few [1]. These structures are diverse morphologically both between families and within plant species [2,3]. They can be pits, deep pockets on the leaf surfaces that extend to the mesophyll, a dense clump of hairs, cavities which extend beneath expanded veins, and rolled vein margins. Sometimes a combination of these types can be found in one species [1,2]. The morphological structure of leaf domatia varies with environmental conditions, both between and within individuals of the same species [3,4]. The size and density of domatia on leaves vary depending on the relative rainfall at the site where the plant occurs [4].

Numerous studies have looked at the role that leaf domatia play in facilitating plant–mite and sometimes plant–ant mutualism [1,5,6,7,8,9]. However, only a handful of studies have focused on the anatomy and ontogeny of leaf domatia [2,3,10,11,12,13,14]. These studies broadly describe the structures and suggest that their formation is characterized by cell differentiation and active cell division in the lower mesophyll. An unpublished study looking at the ontogeny of leaf domatia in four species (*Psychotria capensis* Vatke, *Gardenia thunbergia*, *Coprosma baueri* Endl., and *Pavetta revoluta* Hochst.) from the Rubiaceae also showed that domatia are already present on leaves from an early developmental stage [14]. Trichomes also develop early on and can be fully developed after only one month. In *P. capensis*, the leaves possessed both domatia and trichomes soon after they emerged from the bud, and the number of trichomes increased as the leaf aged. All the domatia of these species increased in number, size and complexity over the life of the leaf [14].

A few studies discuss the anatomical characteristics of domatia, and many of them describe ant-domatia [15,16,17]. To date, only a single comparative study has thoroughly examined the anatomy and development of the different types of domatia. One of these studies looked at the different types of domatia found in *Cinnamomum camphora* (L.) J. Presl., which produces four domatia types (pouch, pubescent pit- or hair-tuft-type, globous pit-type, and dish-type or cavities) within a single leaf. The study revealed that the upper mesophyll tissue, the lower mesophyll tissues, and the tissues filling the rim opening make up the anatomy of the domatia types of this plant. Differences between the domatia types were recognized because of differences in the cell types of the upper mesophyll tissue and the size and number of cells of the rim tissue [3]. This is the only study that provides a better understanding of the modifications that occur at a cellular level when plants are developing domatia.

Additional studies are thus needed for us to gain a better understanding of how plants benefit from producing these structures and how they facilitate the relationship between plants and mites. This is particularly important as some studies suggest that domatia allow plants to communicate with mites, and that they may be providing mites with food in the form of metabolites through specialized structures inside the domatia [5,15]. This suggestion comes from a study looking at the domatia anatomy of *Plectroniella armata* (K.Schum.) Robyns. [15], which showed the presence of channel-like structures made up of thick cuticular folds inside the domatia. The authors believe that these structures assist in the communication between the domatia of this plant and their inhabitants, as well as in the translocation of compounds and metabolites. This further highlight the need for more anatomical studies to explore the presence of the above-mentioned the structure inside of leaf domatia.

The current study describes the leaf anatomy of three plant species with different domatia types belonging to the Rubiaceae. Similar to the study by [15], we also examined the epidermal structure of these species to confirm the presence of modifications in their cuticles.

## 2. Results

### 2.1. The Morphology of Domatia Based on Light Microscopy

The studied plant species possessed different types of domatia. Under a dissecting microscope, the domatia of *R. capensis* and *G. thunbergia* appeared similar (pouch-/pit-type domatia with trichomes). On the other hand, *R. globosa* has a hair-tuft-type domatia with a shallow invagination. The light microscope images reveal the internal anatomical structure of these domatia types (Figure 2, Figure 3 and Figure 4). Compared to the non-domatial tissue (Figure 2a and Figure 3a), the pouch-type domatia of *R. capensis* and *G. thunbergia* (Figure 1 and Figure 2b) comprise five histological parts: the upper epidermis, made up of a single layer of upper epidermal cells; two layers of palisade cells and three tightly packed layers of spongy mesophyll cells; domatial tissue labeled LD; and a single layer of the lower epidermis. The rim tissue consisted of collenchyma cells in the lower epidermal layer with trichomes. *Rothmannia capensis* (Figure 3) had four layers of spongy mesophyll, and these cells were also tightly packed.

The domatial and non-domatial anatomy of *R. globosa* did not differ, and no modifications or cell differentiations were observed in the hair-tuft-type domatia (Figure 4). The domatial tissue consisted of a single layer of epidermal cells, one layer of palisade mesophyll, three layers of spongy mesophyll, and the lower epidermis with simple trichomes (Figure 4b).

### 2.2. Transmission Electron Microscopy Study of the Anatomical Modification of the Epidermis and Cuticle Found inside Leaf Domatia

The results from the TEM study revealed that the plants did possess cuticular folds in the epidermis of the leaf domatia (Figure 5, Figure 6, Figure 7 and Figure 8). In *R. capensis* (Figure 5), the cuticular folds were found consistently inside the domatia (Figure 5), but cuticular folds were also observed in some areas on the adaxial side of the leaf (Figure 6). These were not as pronounced as in the domatial area. Similarly, cuticular folds were present in the domatia of *G. thunbergia* (Figure 7b) and *R. globosa* (Figure 8) but were not regularly seen.

## 3. Discussion

### 3.1. Differences between Domatia Types

This study provides insight into the anatomical differences in the structure of the different domatia types and plant species studied. The major differences in the anatomy of the domatia compared to that of the non-domatial lamina in *G. thunbergia* and *R. globosa* were either one or more of the following: (1) the presence of an invagination on the lower surface, (2) the presence of simple and branched trichomes, (3) the tightly compact parenchyma cells which made up the domatial tissue, (4) the collenchyma cells which formed the rim tissue and in some cases, (5) and a cuticular layer. Apart from the presence of trichomes, there were no differences between the domatial and non-domatial anatomy of *R. globosa*. In a similar study [3], it is suggested that the invagination forms as a result of differences in cell growth rates between the upper and lower regions of the leaf lamina, and that the lower epidermis and lower mesophyll divide more rapidly compared to the upper mesophyll and epidermis, resulting in the formation of the cavity. Similarly, when looking at the base of the domatia or the edge where the pocket starts, we observed tightly packed cells of different sizes, and we speculate that in these plants, also, domatia form as a result of rapidly dividing cells in the domatial region of the leaf.

Trichomes were present in all the plants sampled, and they covered the entire domatia. This might be a family-specific feature, as all three of the plants sampled belong to the Rubiaceae. Studies examining the ontogeny of leaf domatia in four species (*Psychotria capensis*, *Gardenia thunbergia*, *Coprosma baueri*, and *Pavetta revoluta*) from the Rubiaceae revealed that trichomes inside domatia develop early and can be fully developed after only one month, and that the number of trichomes increases with leaf age. [10,13,18]).

Domatial tissue was a common feature in both of the pouch-type domatia, comprising about three layers of tightly packed rectangular cells. The domatial tissue was very distinct from the non-domatial lamina in both plants. This finding echoes that of [19] and others ([3,15]), who also found that the domatia were characterized by modified tissue that may not be found anywhere else in the leaf lamina. These observations suggest that this feature might be universal in plants that possess these domatia types. The rim tissues of both of the pouch-type domatia consisted of collenchyma cells. A study by [13] reported a thick cuticle in the domatia of *Rudgea eugenioides* Standl. (also of the Rubiaceae). In comparison, the hair-tuft-type domatia lacked domatial tissue and rim tissue, and the domatia were marked by the presence of highly abundant trichomes.

### 3.2. Cuticular Folds: Are They Evidence for Plant–Mite Communication?

A study by [15] found that the cuticle of the domatia of *Plectroniella armata* (Rubiaceae) had pronounced and regular folding inside. Within these folds were electron-dense non-cellulosic branching fibrils, extending across the cuticle towards the cavity of the domatia. These channel-like structures and folds are thought to be an indication that some form of communication may be possible between domatia and mites [15]. From our results, we observed similar structures inside and outside of the domatial cuticle of the three species we studied. In *R. capensis* (also Rubiaceae), in particular, the cuticular fold was found consistently inside the domatia and was very pronounced on the underside of the leaf. Unlike in *P. armata*, these cuticular folds were not restricted to the domatial area but were also seen in some areas on the adaxial side of the leaf. However, these folds on the adaxial side were not as regular and pronounced as in the domatial area. Because these folds are present also outside the domatia, it is less likely that these structures form part of a specialized domatial system involved in plant–mite communication. In *G. thunbergia* and *R. globosa*, the cuticle of the domatia was seen to be folded in some areas, but not in other areas. The fact that these features were not regularly seen in the domatia of these plants further suggests that these features are not an inherent part of the domatia.

## 4. Materials and Methods

Three plant species, namely *Gardenia thunbergia* Thunb., *Rothmannia capensis* Thunb., and *Rothmannia globosa* (Hochst.) Keay from Rubiaceae, were selected for this study. The specimens from these plants were examined using Light Microscopy, Scanning Electron Microscopy, and Transmission Electron Microscopy. For each species, a maximum of four leaves were collected at the University of Pretoria botanical gardens.

### 4.1. Preparation for Light Microscopy

From each specimen, a small section of the leaf containing the domatia and surrounding tissue was cut from the midrib and fixed in Formaldehyde, Acetic acid, and Alcohol (FAA) for a week. A series of alcohol solutions were used to dehydrate the leaves, starting with 30% butanol. Daily changes of 50%, 70%, 90%, and then 100% n-butanol were made, and the alcohol was then replaced with paraplast wax. The leaves were embedded in the wax to provide support during sectioning. Serial sections of about 8 µm were cut from the leaves using an ultra-microtome. To view the cell wall and cytoplasm, the sections were stained with safranin and fast green. Permanent mounts were produced using Entellan^®^ (Product 7961, E. Merck, Darmstadt, Germany) rapid mounting medium. The sections were examined using an Olympus Light Microscope, and photographs of the modified regions of domatial epidermis were taken with a camera at a fixed magnification.

### 4.2. Preparation for Transmission Electron Microscopy

Fresh leaf sections were prepared for Transmission Electron Microscopy (TEM) to study the epidermis and look for evidence of cuticular folds inside the leaf domatia. These samples were fixed in a Glutaraldehyde/Formaldehyde (1:1 ratio) solution and dehydrated in an ethanol series in the same way as mentioned above. After that, the leaf material was embedded in LR white resin solution in ethanol. Embedding was a stepwise process, and the resin solution was increased in concentration (20%, 40%, 60%, 80%, and 100%) every hour. The plant material was transferred into gelatin capsules, fresh 100% resin was added, and they were placed in the oven to polymerize for 36 h. The resin block was then trimmed using a surgical blade, sections 0.5 µm thick were cut using a rotation microtome razor, and the contrast of the sections was enhanced with uranyl acetate and lead citrate. These were then viewed in TEM and photographs were taken.

## 5. Conclusions

This study provides useful insights into the anatomy of leaf domatia of three species of the economically important family Rubiaceae. We found that the three study species had distinct differences in their anatomy and that the key features that distinguishes domatia in these species were the presence of an extra layer of tissue in the lower epidermis, the presence of an invagination, and trichomes. Furthermore, the TEM study was in agreement with the results obtained by ([3,15]) and showed that domatia have thick cuticular folds. However, these were not limited to the domatial region, and we thus disagree with the hypothesis that these structures secrete substances that the mites feed on. These structures cannot be linked with, and do not form part of, a plant–mite communication. However, given we that have only examined three species within the very large and diverse Rubiaceae, considerably more effort is needed to adequately document the developmental anatomy and micromorphology of leaf domatia in other plant families.

## Figures and Tables

**Figure 1 plants-11-03126-f001:**
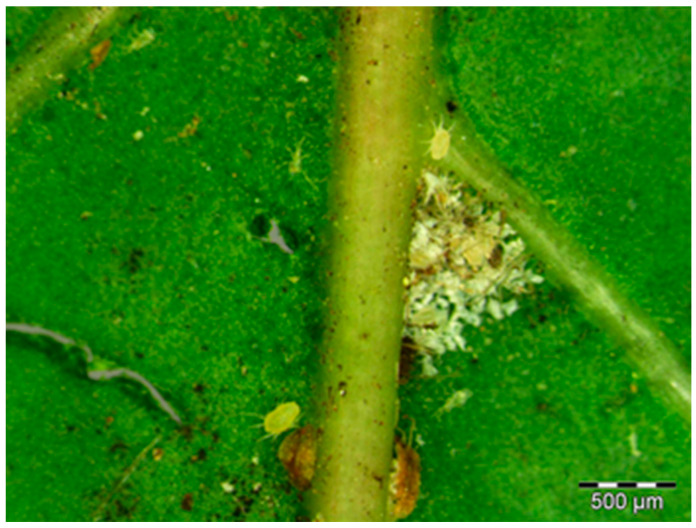
Stereomicrograph showing the location of the leaf domatium of *Rothmannia globasa* (Hochst.) Keay Rubiaceae in the adaxial side of the leaf.

**Figure 2 plants-11-03126-f002:**
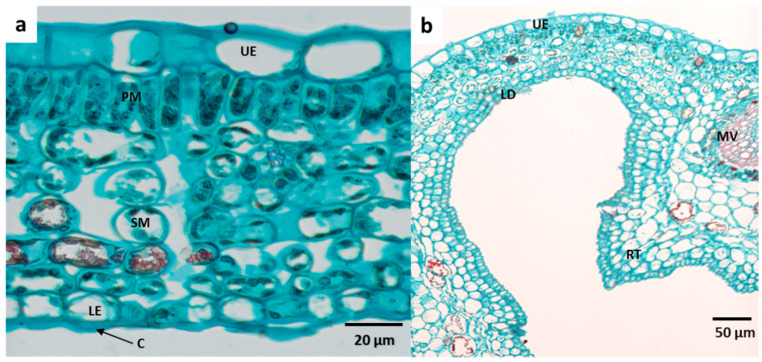
Comparison of domatial and non-domatial tissue of the pouch-type domatia of *Rothmannia capensis*; (**a**) non-domatial area of leaf blade, (**b**) domatial tissue with tightly packed cells. C: cuticle, LD: leaf domatial tissue, LE: lower epidermis, MV: mid-vein, UE: upper epidermis, PM: palisade mesophyll layer, RT: rim tissue, and SM: spongy mesophyll layer.

**Figure 3 plants-11-03126-f003:**
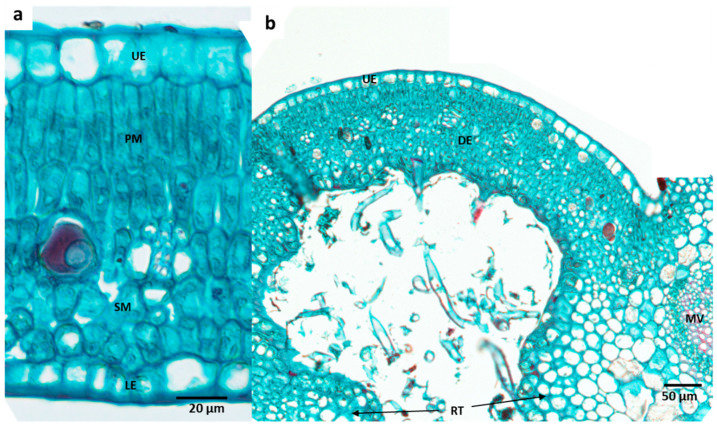
Comparison of non-domatial with domatial tissue of the pouch-type domatia of *Gardenia thunbergia*; (**a**) non-domatial area, (**b**) domatial tissue with trichomes. LD: leaf domatial tissue, LE: lower epidermis, MV: mid-vein, UE: upper epidermis, PM: palisade mesophyll layer, RT: rim tissue, SM: spongy mesophyll layer, and T: trichomes.

**Figure 4 plants-11-03126-f004:**
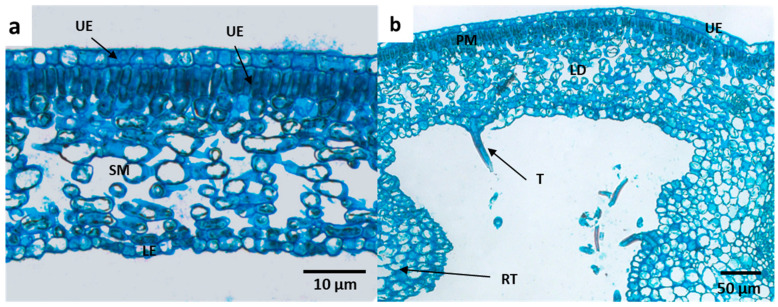
Comparison of domatial and non-domatial tissue of the hair-tuft-type domatia of *Rothmannia globosa*; (**a**) non-domatial area, (**b**) domatial opening with trichomes. LD: leaf domatial tissue, LE: lower epidermis, MV: mid-vin, UE: upper epidermis, PM: palisade mesophyll layer, RT: rim tissue, SM: spongy mesophyll, and T: trichomes.

**Figure 5 plants-11-03126-f005:**
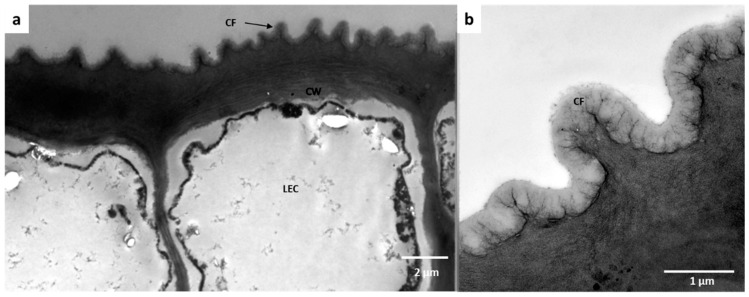
The anatomical modification of the epidermis found in the domatia of *Rothmannia capensis*; (**a**) cuticular fold present in the epidermis of the domatia, (**b**) close-up of the cuticular folds found inside the domatia. CF: cuticle fold, CW: cell wall, and LEC: lower epidermal cell.

**Figure 6 plants-11-03126-f006:**
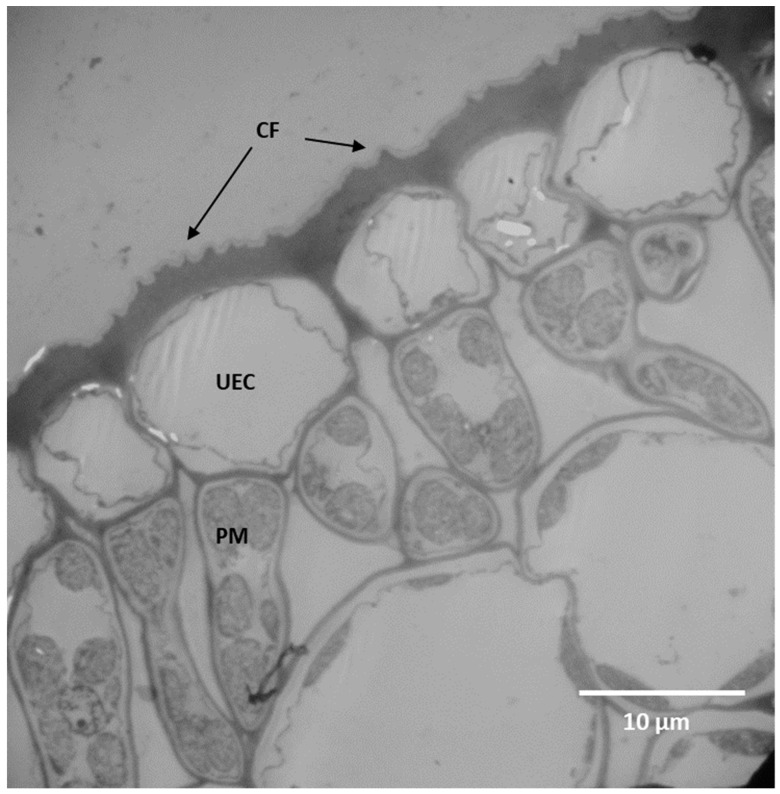
The anatomical modification of the epidermis in the adaxial side of the leaf in *Rothmannia capensis*. CF: cuticle fold, PM: palisade mesophyll cell, UEC: upper epidermal cell wall, and LE: lower epidermal cell.

**Figure 7 plants-11-03126-f007:**
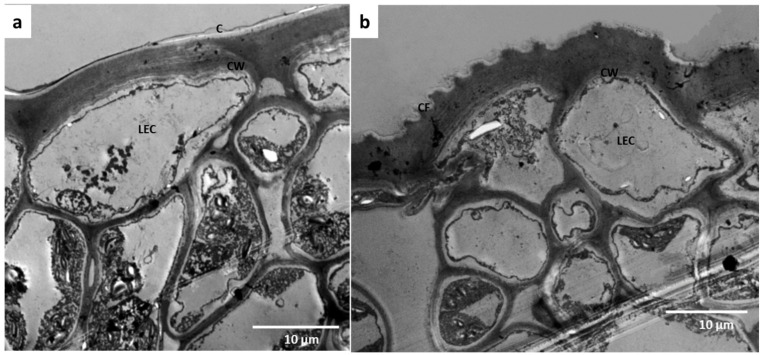
The anatomical modification of the epidermis in the domatia of *Gardenia thunbergia*; (**a**) non-domatial epidermis, (**b**) cuticular fold in domatial epidermal layer. C: cuticle, CF: cuticular fold, CW: cell wall, and LEC: lower epidermal cell.

**Figure 8 plants-11-03126-f008:**
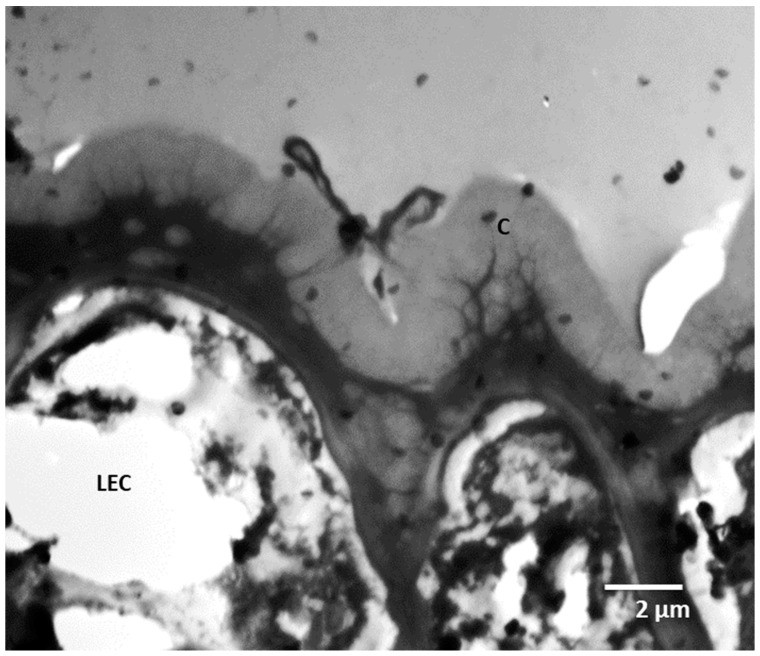
The anatomical modification of the epidermis in the domatia of *Rothmannia globose.* C: cuticle and LEC: lower epidermal cell.

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
