# Peer review of "A Comparative Study of the Anatomy of Leaf Domatia in Gardenia thunbergia Thunb., Rothmannia capensis Thunb., and Rothmannia globosa (Hochst.) Keay (Rubiaceae)"

_plants, 2022, doi:10.3390/plants11223126_

Round 1

Reviewer 1 Report

Journal Plants (ISSN 2223-7747)

Manuscript ID plants-1931354

Type Communication

Title A comparative study of the anatomy of leaf domatia in six woody plant species

Dear Editor,

Thank you for inviting me to review the manuscript “ A comparative study of the anatomy of leaf domatia in six woody plant species”.

The manuscript requires substantial revision and additional information.

Below are some comments that will improve the quality of this manuscript.

Title

1. Please in the title of the manuscript give the names of plant species

Abstract

2. The abstract should include the following elements: (a) place the introduction addressed in a broad context, (b) highlight the purpose of the study, (c) describe briefly the main methods applied, (d) summarize the main findings, (e) and indicate the main conclusions.

Key words

3.  Eliminate phrases appearing in the title of the manuscript.

Introduction

4.  The Introduction should be re-edited to give it a more scientific character; while citing 12 literature references, the authors write two sentences without providing scientifically relevant information.

„ Numerous studies have looked at the role that leaf domatia play to facilitate the 36 plants-mites and sometimes plant-ant mutualism [1]; [4]; [5]; [6]; [7]; [8]. However, only a 37 handful of studies have focused on the anatomy and ontogeny of leaf domatia [2]; [3]; [9]; 38 [10]; [11]; [12].

5. The manuscript should be completed with citations of the latest scientific literature related to the issue addressed in the study.

6. The specific aim of the undertaken research should be justified and formulated clearly.

Correct the literature reference “plant-ant utualizm [1]; [4]; [5]; [6]; [7]; [8]”.

Results

7. Figure 1a,d; figures 3-5;  – the poor quality of the photographs should be improved.

8. Specify what information the authors want to convey to the reader by presenting the figures.

9. The description of the figures requires editing in line with the guidelines from the PLANTS journal.

10. In the description of the figures, eliminate e.g. “... at high magnification ...”, “... is a close-up image...”.

11. The layout of the photos is careless; the structure or organelles (TEM) cannot be seen clearly.

12. Do the Authors mean protective or secretory trichomes? Provide the documentation of these trichomes.

13. The Results include a text containing 308 words; the text requires addition of scientific information and providing of complete research results (as in publications in journals with an international range and a high impact factor, e.g. PLANTS).

Discussion

14. The Discussion should be re-edited step by step. First, each trait should be presented as analyses in the authors’ study and then it should be discussed in comparison with findings reported by other authors.

15. Subsection 3.2. presents no discussion.

Materials and Methods

16. Complete the microscopic methodology with detailed information on material fixation - concentrations, cutting the material, preparation of semi-thin and ultra-thin slides, staining, contrasting, etc.

17. Check the thickness in “Serial sections of about 8 μm were...”

Conclusion

18. The conclusions should be a response to the aim of the study and assumed theses.

19. Please formulate specific conclusions following from the research

References

20. Check the list of references and the entire text so that they follow the guidelines for authors.

21. Latin names of the species should be written in italics l. l. 248, 259 e.t.c.

Reviewer 2 Report

The manuscript presents the anatomical description of leaf domatia which are associated with the mutualistic relations between plants and predacious mites. Despite this is interesting  investigation of domatia anatomy in three different plant species I have major questions to this work: 

1. I do not agree with the statement in Introduction that only few studies (and even only a single study – L.42) are dealing with anatomical structure of leaf domatia. I am sure that many studies to some extent are focused on domatial morphology and anatomy. Here are only two examples of such studies:

 - Romero M.F., Salas R., Gonzalez A.M. Morpho-anatomic studies of leaf domatia in Argentinian rubiaceae [Estudios morfo-anatómicos de domacios foliares en rubiáceas Argentinas] (2015) Boletin de la Sociedad Argentina de Botanica, 50 (4), pp. 493 - 514

- Review paper: Schmidt R.A. Leaf structures affect predatory mites (Acari: Phytoseiidae) and biological control: a review. Exp Appl Acarol (2014) 62:1–17

I think authors should extend their Introduction including more relevant studies.

 2. Results on comparative analysis of domatia in three different Rubiaceae species are presented only descriptively without any measurements. To correctly answer their questions authors should increase the number of studied leaves (four leaves are too few for quantitative analysis). Conclusion on “thickened cuticle” should be confirmed by measurement of cuticule thickness but not only by visual observation and simple description.  It would be properly to count the number of trichoms and the number of cuticular bonds inside the domatia to compare different plant species on the base of real quantitative data. It worth also to apply statistical analysis before conclusions.  

I recommend to authors to continue the investigation and to prepare a new manuscript.

Reviewer 3 Report

I found this manuscript quite interesting and very well prepared as are the figures.  I only wish all manuscripts I review would be of this calibre. As far as I am, concerned it is ready to be published.

Author Response

Thank you so much for taking to to review my manuscript and for the praises. 

Round 2

Reviewer 1 Report

Plants (ISSN 2223-7747)

Manuscript ID plants-1931354

Type Communication

Title A comparative study of the anatomy of leaf domatia in six woody plant species

Authors Sivuyisiwe Situngu * , Nigel Barker

Section Plant Development and Morphogenesis

Special Issue Functional Plant Anatomy – Structure, Function and Environment

Dear Editor

I regret to say that the figures in manuscript ID plants-1984103 have still not been corrected.

General comments

1.      It is assumed that photographs in a figure are arranged from the smallest to the largest magnification (which is not the case in the reviewed manuscript)

2.      Usually, when arranging photographs, the elements should be positioned correctly, e.g. the upper epidermis of leaves upwards; this is not the case in the manuscript.

3.      Bars should be arranged in an esthetic way, and photos should be placed at an appropriate distance, unlike in the reviewed manuscript.

4.      Photos should be properly arranged and empty photos (not presenting anything) should be cut out

Detailed comments

1.      Figure 1. a-c, 2b -d, 3b  – the contrast should be corrected.

2.      Fig. 1b, 2b, 3b, 3c – the illegible fragment should be cut off and enlarged to show the anatomical structure.

3.      Figure 4A -E the photographs in this figure should be properly arranged (the current arrangement is unacceptable - see the comments above).

4.      Fig.4c, e – The description should explain what the authors want to present.

5.      fig. 4d – the contrast should be corrected.

6.      Figure 5A and 6a – very bad quality (contrast); it is unacceptable in its present form.

7.      Fig. 5a, 6b – the photographs should be enlarged and the sharpness should be improved, description should indicate what the authors want to present, and the missing markings should be added.

8.      The discussion is not a place to cite the figures of the present study (page 6 and 7)

9.      Check the exact citations (l. 44,  92, 126, 162 etc.).

Taking into account the international scope of the journal and the high requirements that authors of other publications have had to meet, I believe that the manuscript in this form should not be processed further for publication.

I leave the final decision to the Editor.

Best regards

Reviewer

Author Response

Dear reviewer,

I welcome your useful comments and I have considered them. 

  1. With regards to the figures, it is unfortunate that I cannot edit them because I do not have the originals due to theft. I played with the contrast for figure 1 and 4 however, this affected the quality of the pictures and I decided to leave them as they are. 
  2. I removed the figure citations in the discussion as you suggested. 
  3. I scrutinised the citation and made correct where there were mistakes. 

Thank you. 

Reviewer 2 Report

I still think that some important questions remain inconclusive in this manuscript. The most important of them are:

1) the presentation of results concerning domatial anatomy remains incorrect. If “this is not a quantitative study” and authors did not measure cuticle thickness, cell sizes and trichomes number, so they CAN NOT conclude about the differences in cuticle thickness, sizes of cells and abundance of trichomes between domatial and non domatial tissues. It is impossible to assess quantitative characteristics only by eye. Such conclusions should be confirmed by measured data and statistical analysis of differences significance. Therefore, authors should exclude from the entire text of the paper such statements as well as all conclusions from these statements:

“thick cuticle” (L.24, L.378)

“enlarged collenchyma cells” (L.160, 161, 286)

“appeared thickened” (L.169)

“a thickened cuticular layer” (L.262)

“tightly packed cells of different sizes” (L.269)

“rapidly dividing cells” (L.270)

“and these cells appeared relatively 280 small in comparison to rim tissue cells” (L.280)

“of highly abundant trichomes” (L.306)

2) The conclusions should correspond to the purpose of the work. Authors specify that “The current study describes the leaf anatomy of three plant species with different domatia types”, however there is a lack of clear outcomes to this purpose in Conclusions. The main part of Conclusions is engaged by description of limitations of this study rather than by conclusions from results. 

Author Response

Dear  reviewer, 

Your comments were very useful and we appreciated them. See our response to them below: 

  1. We have revised the way the results are presented and the discussion section and appreciate the reviewer for making us aware that we can’t make such statement if the study is not quantitative.

  2. The conclusion has been edited thoroughly to summerise the outcomes of the study

 Thank you. 

Round 3

Reviewer 2 Report

The authors improved the ms according to reviewer comments. 

Author Response

I have made all the changes requested in round 2. thank you so much!